

# Design of a robust active fuzzy parallel distributed compensation anti-vibration controller for a hand-glove system

Leila Rajabpour[1], Hazlina Selamat[1], Alireza Barzegar[2] and Mohamad Fadzli Haniff[3]

[1] School of Electrical Engineering, Universiti Teknologi Malaysia, Johor Bahru, Johor, Malaysia
[2] Electrical Engineering, Nanyang Technological University, Singapore
[3] Malaysia-Japan International Institute of Technology, Universiti Teknologi Malaysia, Kuala Lumpur, Federal Territory, Malaysia

## ABSTRACT

Undesirable vibrations resulting from the use of vibrating hand-held tools decrease the tool performance and user productivity. In addition, prolonged exposure to the vibration can cause ergonomic injuries known as the hand-arm vibration syndrome (HVAS). Therefore, it is very important to design a vibration suppression mechanism that can isolate or suppress the vibration transmission to the users' hands to protect them from HAVS. While viscoelastic materials in anti-vibration gloves are used as the passive control approach, an active vibration control has shown to be more effective but requires the use of sensors, actuators and controllers. In this paper, the design of a controller for an anti-vibration glove is presented. The aim is to keep the level of vibrations transferred from the tool to the hands within a healthy zone. The paper also describes the formulation of the hand-glove system's mathematical model and the design of a fuzzy parallel distributed compensation (PDC) controller that can cater for different hand masses. The performances of the proposed controller are evaluated through simulations and the results are benchmarked with two other active vibration control techniques-proportional integral derivative (PID) controller and active force controller (AFC). The simulation results show a superior performance of the proposed controller over the benchmark controllers. The designed PDC controller is able to suppress the vibration transferred to the user's hand 93% and 85% better than the PID controller and the AFC, respectively.

# INTRODUCTION

Vibrations of the hand-held tools can cause harmful health conditions among the operators. There have been many investigations on the effects of the long-term vibration exposure on the human body (*Shen & House, 2017*) and many damages have been reported including HAVS, muscle weakness, white finger, loss of grip strength, sensory nerve damage, and muscle and joint injuries in the hand and arm (*Gerhardsson et al., 2020*; *Vihlborg et al., 2017*). The effects of anti-vibration (AV) gloves on reducing the health risks

Corresponding author
Hazlina Selamat, hazlina@utm.my

of the vibrating tools have been investigated in many studies, and various types of anti-vibration gloves have been introduced to the market such as gel-filled, air-filled and leather AV gloves (*Hamouda et al., 2017*).

Various biodynamic models of the hand have been proposed in International Standard ISO 10068:2012 (*Dong et al., 2009*; *Rezali & Griffin, 2018*) to investigate the transmissibility of the vibration to the hand. The strategy of building the hand model is based on introducing the simple mechanical damper and spring elements, which are arranged in series or parallel depending on the approach used such as Maxwell or Kelvin models of muscles (*Jones, 2001*), to represent the viscoelastic characteristics of the hand soft tissues. The factors such as types of tools used, the points at which vibrations are transmitted (*e.g.* the hand or the arm) and the required levels of accuracy determine the modeling approach to be used. For example, in *Mazlan & Ripin (2015)* a two degree of freedom (DOF) model was adopted while some higher DOF models can be found in *Kamalakar & Mitra (2018)*. A precise representation of different impulsive tools and their working postures and the directions of the applied hand force is illustrated in *Dong et al. (2015)*.

Incorporating a glove into the hand system requires additional mechanical damper and spring elements in the equivalent mechanical system. For example, the model proposed by *Dong et al. (2009)* is a 7-DOF model that represents the viscous, elastic and inertial properties of the glove by the equivalent springs and dampers. Adding an active element to the passive anti-vibration glove can improve the efficacy of the system. In active vibration control (AVC), an actuator is utilized to apply an external force or displacement based on the measurement of the system response through feedback control (*Preumont, 2018*). The acceleration, displacement and velocity measurements from the sensors are used by the control system to provide control signals for actuators based on the chosen control strategy.

The design of the control scheme could be quite challenging in this area since many parameters are affecting the performance of the controller such as sensor and actuator's fault (*Gao & Liu, 2020*; *Tahoun, 2020*) and uncertainties in the system parameters (*Chen et al., 2019*; *Li et al., 2017*; *Tahoun, 2017*). Besides, the vibration of different body parts occurs at different frequencies and therefore would add complexity to the control design process (*Hassan et al., 2010*).

Various types of control schemes have been developed and employed in AVC structures. Each new control method has been designed for a specific system with a specific dynamics, such as integral-based controllers in *Zuo & Wong (2016)*, sliding mode controls (*Hamzah et al., 2012*; *Lin, Su & Li, 2019*), PID control (*Rani et al., 2011*), AFC (*Theik & Mazlan, 2020*) and artificial intelligent-based control strategies including fuzzy and neural networks (*Kalaivani, Sudhagar & Lakshmi, 2016*). In *Lekshmi & Ramachandran (2019)* a Genetic Algorithm-optimized PID controller is proposed to suppress Parkinson's Tremor. In *Liu et al. (2018)*, an adaptive neural network controller is proposed to control the suspension systems, and some fuzzy active suspension system controllers are also proposed in *Sun et al. (2018)* and *Tabatabaei et al. (2010)*.

The AFC proposed in *Zain, Mailah & Priyandoko (2008)* is an effective method designed to carry out the robust control of dynamic systems in the presence of disturbance

and uncertainties. It is proved that in AFC, the stability and robustness of the system remain unchanged by compensation action of the control strategy when a number of disturbances are applied to the system (*Mailah et al., 2009*).

The research on fuzzy systems and control has resulted in the development of many controller designs especially by using the Takagi-Sugeno (T-S) fuzzy model as an effective and simple tool in modeling and control of nonlinear systems or systems with variable parameters (*Rajabpour, Shasadeghi & Barzegar, 2019*; *Sadeghi, Vafamand & Khooban, 2016*). Additionally, the T-S model can generate an exact representation of the nonlinear system by a set of linear subsystems. Since the hand mass is variable for different users in the hand-glove system, we propose to use the fuzzy T-S model for the standard range of various hand masses and employing the idea of fuzzy parallel distributed compensation (PDC) technique, proposed by *Wang, Tanaka & Griffin (1995)*, to control the fuzzy T-S modeled systems by designing a linear feedback controller for each linear subsystem.

Originally, the idea of PDC has been applied to control nonlinear systems by linearizing them around different operation points. Here, we used this idea to control the linear hand-glove system but we used the ability of PDC control to make the model and controller more flexible for different users with different hand masses which has not been considered in the previous controller designs. Building on the anti-vibration gloves literature, we propose the design of an active anti-vibration controller for the hand-glove system to suppress the transmitted vibration to the hand while using hand-held tools. To our knowledge, it is the first anti-vibration glove with an active controller. In this paper, the mathematical model of the hand-glove system is formulated and three different controllers- a PID controller, an AFC and a fuzzy PDC controller- are designed and applied to the hand-glove system. The performances of the three controllers are then compared.

To summarize, the key contributions of this paper are twofold: the T-S fuzzy modeling of the hand-glove system for variable hand mass parameter and the design of an active anti-vibration glove based on the fuzzy PDC controller, which is robust to variation in hand masses, making the active anti-vibration glove suitable for different users.

This paper is organized as follows: the "Introduction" presents the mathematical formulation of the hand-glove model. "ACtive-Vibration Controller (AVC) Design" describes the development of the fuzzy PDC active anti-vibration controller for the hand-glove system. "Simulation Results" demonstrates the simulation results and analysis and "Conclusion" concludes the paper.

## MODELLING OF ACTIVE HAND-GLOVE SYSTEM

To represent the dynamic behavior of the hand-glove system under the influence of vibration, the system is represented by a three-degree of freedom model as represented in Fig. 1. All the masses are assumed to be rigid and the mass of each segment is considered to be a percentage of the total body mass (*Rezali & Griffin, 2018*). Since our main focus is on the transmitted vibration to the hand and to reduce the complexity of the model, the palm and fingers are considered to be one body segment and the palm is

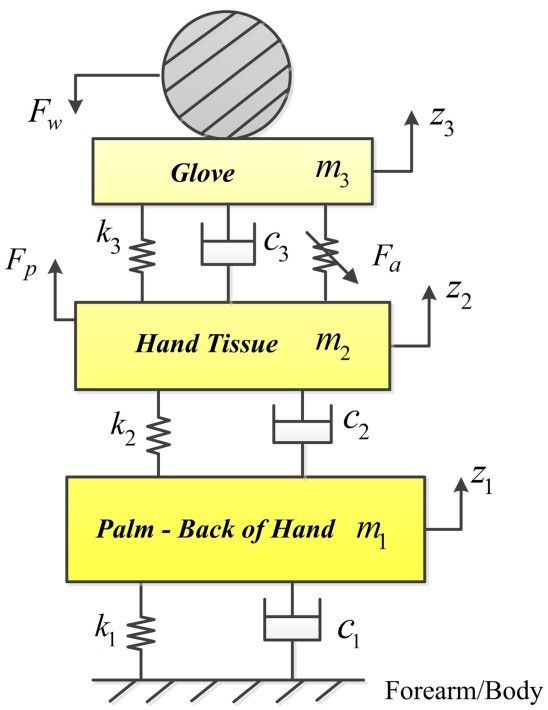

**Figure 1  The three DOF hand-glove model.**

**Table 1  Gloved-hand model parameters.**

| Symbol | Description | Unit |
|---|---|---|
| $m_1$ | Mass of the palm and back of hand | $kg$ |
| $m_2$ | Mass of the hand tissue | $kg$ |
| $m_3$ | Mass of the glove | $kg$ |
| $c_1$ | Stiffness coefficient of the hand | $N/m$ |
| $c_2$ | Stiffness coefficient properties of the soft tissue of the hand | $N/m$ |
| $c_3$ | Stiffness coefficient properties of the glove material | $N/m$ |
| $k_1$ | Damping coefficient of the hand | $N.s/m$ |
| $k_2$ | Damping coefficient of the soft tissue of the hand | $N.s/m$ |
| $k_3$ | Damping coefficient of the glove material | $N.s/m$ |
| $z_i$ | Displacement of mass $m_i$ | $m$ |
| $F_a$ | Control Input (*Actuator force*) | $N$ |
| $F_w$ | Vibration Input (*Disturbance*) | $m/a^2$ |
| $F_p$ | Push force | $N$ |

assumed to be lying on the vibration surface. Table 1 lists the symbols, units and descriptions of the parameters of the hand-glove system shown in Fig. 1.

## The equations of motion of the active hand-glove model

In Fig. 1, since the actuator is assumed to be placed on the palm under the glove, the actuator force $F_a$ is applied between the soft tissue of the hand, mass $m_2$ and glove material with mass $m_3$. The hand pushing the tool handle is represented by $F_p$.

The dynamic equations of motion of the system can be written as follows:

$$m_1(t)\ddot{z}_1 + (c_1 + c_2)\dot{z}_1 - c_2\dot{z}_2 + (k_1 + k_2)z_1 - k_2z_2 = 0 \tag{1a}$$

$$m_2\ddot{z}_2 - c_2\dot{z}_1 + (c_2 + c_3)\dot{z}_2 - c_3\dot{z}_3 - k_2z_1 + (k_2 + k3)z_2 - k_3z_3 = F_a - F_p \tag{1b}$$

$$m_3\ddot{z}_3 - c_3\dot{z}_2 + c_3\dot{z}_3 - k_3z_2 + k_3z_3 = -F_a + F_w \tag{1c}$$

where $z_i$, $\dot{z}_i$ and $\ddot{z}_i$ represent the displacement, velocity and acceleration of mass $m_i$; $F_a$ is the actuator force, $F_p$ is the push force and $F_w$ is the input vibration or disturbance to the system that needs to be controlled.

In this model, the mass of the hand is assumed to be non-constant, $m_1$ $(t)$, i.e. the model can be used to represent the hand system for different users. The Eqs. (1), (1b) and (1c) can be arranged in the following form:

$$\ddot{z}_1 = -\frac{c_1 + c_2}{m_1(t)}\dot{z}_1 + \frac{c_2}{m_1(t)}\dot{z}_2 - \frac{k_1 + k_2}{m_1(t)}z_1 + \frac{k_2}{m_1(t)}z_2 \tag{2a}$$

$$\ddot{z}_2 = \frac{c_2}{m_2}\dot{z}_1 - \frac{c_2 + c_3}{m_2}\dot{z}_2 + \frac{c_3}{m_2}\dot{z}_3 + \frac{k_2}{m_2}z_1 - \frac{k_2 + k_3}{m_2}z_2 + \frac{k_3}{m_2}z_3 + \frac{F_a}{m_2} - \frac{F_p}{m_2} \tag{2b}$$

$$\ddot{z}_3 = \frac{c_3}{m_3}\dot{z}_2 - \frac{c_3}{m_3}\dot{z}_3 + \frac{k_3}{m_3}z_2 - \frac{k_3}{m_3}z_3 - \frac{F_a}{m_3} + \frac{F_w}{m_3} \tag{2c}$$

Considering $x_1 = z_1$, $x_2 = z_2$, $x_3 = z_3$, $x_4 = \dot{z}_1$, $x_5 = \dot{z}_2$ and $x_6 = \dot{z}_3$ for each of the state variables in Eqs. (2a), (2b) and (2c), it can be rewritten in the state-space equation given by Eq. (3).

$$\dot{\mathbf{x}}(t) = \mathbf{A}(t)\mathbf{x}(t) + \mathbf{B}(t)\mathbf{u}(t) + \mathbf{E}(t)\mathbf{w}(t)$$
$$\mathbf{y}(t) = \mathbf{C}\mathbf{x}(t) \tag{3}$$

where matrices $\mathbf{A}$, $\mathbf{B}$, $\mathbf{C}$ and $\mathbf{E}$ are the system matrix, the input matrix, output matrix and the disturbance matrix, respectively. The hand vibration, $\dot{x}_4$ or $\ddot{z}_1$ is considered as the performance output reference. Equation (3) can be further written in the following form:

$$
\begin{bmatrix} \dot{x}_1 \\ \dot{x}_2 \\ \dot{x}_3 \\ \dot{x}_4 \\ \dot{x}_5 \\ \dot{x}_6 \end{bmatrix} =
\begin{bmatrix}
0 & 0 & 0 & 1 & 0 & 0 \\
0 & 0 & 0 & 0 & 1 & 0 \\
0 & 0 & 0 & 0 & 0 & 1 \\
-\dfrac{k_1 + k_2}{m_1(t)} & \dfrac{k_2}{m_1(t)} & 0 & -\dfrac{c_1 + c_2}{m_1(t)} & \dfrac{c_2}{m_1(t)} & 0 \\
\dfrac{k_2}{m_2} & -\dfrac{k_2 + k_3}{m_2} & \dfrac{k_3}{m_2} & \dfrac{c_2}{m_2} & -\dfrac{c_2 + c_3}{m_2} & \dfrac{c_3}{m_2} \\
0 & \dfrac{k_3}{m_3} & -\dfrac{k_3}{m_3} & 0 & \dfrac{c_3}{m_3} & -\dfrac{c_3}{m_3}
\end{bmatrix}
\begin{bmatrix} x_1 \\ x_2 \\ x_3 \\ x_4 \\ x_5 \\ x_6 \end{bmatrix}
$$

$$
+ \begin{bmatrix} 0 & 0 \\ 0 & 0 \\ 0 & 0 \\ 0 & 0 \\ \dfrac{1}{m_2} & -\dfrac{1}{m_2} \\ -\dfrac{1}{m_3} & 0 \end{bmatrix}
\begin{bmatrix} F_a \\ F_p \end{bmatrix} +
\begin{bmatrix} 0 \\ 0 \\ 0 \\ 0 \\ 0 \\ \dfrac{1}{m_3} \end{bmatrix} F_w \tag{4}
$$

$$y(t) = \begin{bmatrix} 0 & 0 & 0 & 1 & 0 & 0 \end{bmatrix} \begin{bmatrix} x_1 & x_2 & x_3 & x_4 & x_5 & x_6 \end{bmatrix}^T \tag{5}$$

Another approach that can be used to model the active hand-glove system is the fuzzy T-S model and is described next.

## Fuzzy Takagi-Sugeno model

Fuzzy logic has the ability to deal with nonlinearities and uncertainties in the system where standard analytical models are usually ineffective. It can be used for both modeling and design of the controller. Most nonlinear systems can be approximated by a T-S fuzzy model. The T-S fuzzy model of a dynamic system with control input and disturbance input can be represented by the following rules:

*Model Rule i:*

**IF** $s_1(t)$ is $M_{i1}$ and $\cdots$ and $s_p(t)$ is $M_{ip}$,

$$\textbf{THEN} \begin{cases} \dot{\mathbf{x}}(t) = \mathbf{A}_i\mathbf{x}(t) + \mathbf{B}_i\mathbf{u}(t) + \mathbf{E}_i\mathbf{w}(t), \\ \mathbf{y}(t) = \mathbf{C}_i\mathbf{x}(t), \end{cases} \quad i = 1, 2, \ldots, r, \tag{6}$$

where $r$ shows the number of rules, $M_{ij}$ denotes the fuzzy set, and $\mathbf{x}(t) \in \mathbf{R}^{n \times n}$, $\mathbf{u}(t) \in \mathbf{R}^{n \times 1}$ and $\mathbf{w}(t) \in \mathbf{R}^{n \times 1}$ are state vectors, controlled input vector and disturbance input vector, respectively. Note that $s_i(t)$ is a hypothetical variable which can be a function of state variables, disturbance inputs, system parameters and/or time.

By considering each set of $(\mathbf{A}_i, \mathbf{B}_i, \mathbf{E}_i, \mathbf{C}_i)$ as a subsystem, the overall output of the fuzzy system can be represented as

$$\dot{\mathbf{x}}(t) = \frac{\sum_{i=1}^{r} \alpha_i(\mathbf{s}(t))(\mathbf{A}_i\mathbf{x}(t) + \mathbf{B}_i\mathbf{u}(t) + \mathbf{E}_i\mathbf{w}(t))}{\sum_{i=1}^{r} \alpha_i(\mathbf{s}(t))} \tag{7a}$$

$$= \sum_{i=1}^{r} h_i(\mathbf{s}(t))(\mathbf{A}_i\mathbf{x}(t) + \mathbf{B}_i\mathbf{u}(t) + \mathbf{E}_i\mathbf{w}(t)),$$

$$\mathbf{y}(t) = \frac{\sum_{i=1}^{r} \alpha_i(\mathbf{s}(t))(\mathbf{C}_i\mathbf{x}(t))}{\sum_{i=1}^{r} \alpha_i(\mathbf{s}(t))} \tag{7b}$$

$$= \sum_{i=1}^{r} h_i(\mathbf{s}(t))(\mathbf{C}_i\mathbf{x}(t)),$$

in which for all $t$

$$\mathbf{s}(t) = [s_1(t) \ s_2(t) \ \ldots \ s_p(t)], \tag{8}$$

$$\alpha_i(\mathbf{s}(t)) = \prod_{j=1}^{v} M_{ij}(s_j(t)) \tag{9}$$

and

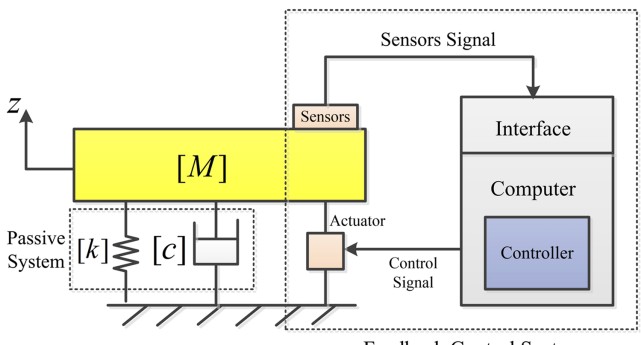

**Figure 2 Active vibration control diagram.**

$$h_i(\mathbf{s}(t)) = \frac{\alpha_i(\mathbf{s}(t))}{\sum\limits_{i=1}^{r} \alpha_i(\mathbf{s}(t))}.$$
(10)

Note that the term $M_{ij}(s_j(t))$ shows the membership grade of $s_j(t)$ in $M_i$. Also, since $\alpha_i(\mathbf{s}(t)) \geq 0$ and $\sum_{i=1}^{r} \alpha_i(\mathbf{s}(t)) \geq 0$ for $i = 1,2,\ldots,r$, then we have $h_i(s(t))) \geq 0$ and $\sum_{i=1}^{r} h_i(s(t)) = 1$ for all $t$.

Next, different types of active vibration control techniques applied to the system to suppress the vibration to the desired level are described.

## ACTIVE-VIBRATION CONTROLLER (AVC) DESIGN

In AVC, the control objective is to attenuate the undesired vibration to the desired level by use of an actuator, which could be a piezoelectric device or an electric motor. The incoming vibration is sensed by using a sensing mechanism and the actuator reacts to these vibrations by producing a cancelation signal. The sensors used in AVC are mainly of piezoelectric type (*Miljković, 2009*). Figure 2 shows the arrangement of the AVC mechanism. The vibration cancelation process can be achieved by using different feedforward and feedback controllers. In this figure, $[M]$ is the mass matrix, matrix $[c]$ contains the system damping coefficients and $[k]$ contains the stiffness coefficients of the system.

The working principle of an AVC system is based on collecting data from sensing devices and then generating a control signal for the actuator. The control signal is generated from the measurements of the displacement, velocity or acceleration of the mass $M$ that is fed back. Based on these measurements, the control signal produced depends on the chosen control strategy or algorithm. The followings explain three different control algorithms applied to the hand-glove system to suppress vibrations transmitted from the tools used to the workers' hands.

### Proportional-integral-derivative (PID) controller

The PID controller to be used in the AVC system is is a standard PID control algorithm given by Eq. (11):

$$u(t) = K_p\, e(t) + K_i \int e(t)dt + K_d\, \dot{e}(t)$$
(11)

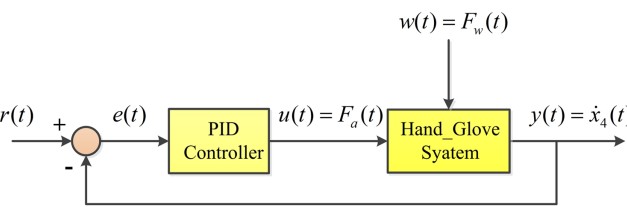

**Figure 3  PID control schematic.**

where $u(t)$ is the control signal, $e(t)$ is the error signal and $K_p$, $K_i$ and $K_d$ are the proportional, integral and derivative gains, respectively. Figure 3 shows the feedback control system employing the PID controller. Equation (11) can be written in the following transfer function form:

$$U(s) = K_p + K_i/s + K_d\,s \qquad (12)$$

The PID gains $K_p$, $K_i$ and $K_d$ need to be tuned, by trial and error tuning method, since we get the desired response.

## Active force controller

Active force control (AFC) proposed in *Zain, Mailah & Priyandoko (2008)* is an effective method designed to carry out the robust control of dynamic systems in the presence of disturbance and uncertainties. It is proved that in AFC, the stability and robustness of the system remain unchanged by compensation action of the control strategy when a number of disturbances are applied to the system (*Abdelmaksoud, Mailah & Abdallah, 2020*; *Gohari & Tahmasebi, 2017*).

The schematic of the AFC controller for a suspension system, which has a similar representation of the hand-glove system, is shown in *Mailah et al. (2009)*.

One practical example of applying AFC to a suspension system to control the unwanted vibration is given in *Mohamad, Mailah & Muhaimin (2006)* and it showed the effectiveness of the applied AFC to cancel the vibration with different disturbances levels.

## Fuzzy parallel distributed compensation (PDC) controller

Fuzzy logic is an effective way to decompose a nonlinear system control into a group of local linear controls based on a set of design-specific model rules. PDC is one of the many varieties of fuzzy logic that can be implemented in control systems. For a given fuzzy T-S framework, the state feedback based on PDC is usually applied. The key idea of the PDC technique is to divide the nonlinear system to some linear subsystems and then design some controllers for each of the linear subsystems and subsequently obtain the overall controller by the fuzzy blending of the local controllers (*Sadeghi, Safarinejadian & Farughian, 2014*; *Sadeghi, Rezaei & Mardaneh, 2015*; *Nguyen, Dambrine & Lauber, 2016*) *i.e.* designing a compensator for each rule of the fuzzy model.

The block diagram of the system with fuzzy PDC controller is illustrated in Fig. 4. As can be seen, the overall PDC controller is applied to the original nonlinear system.

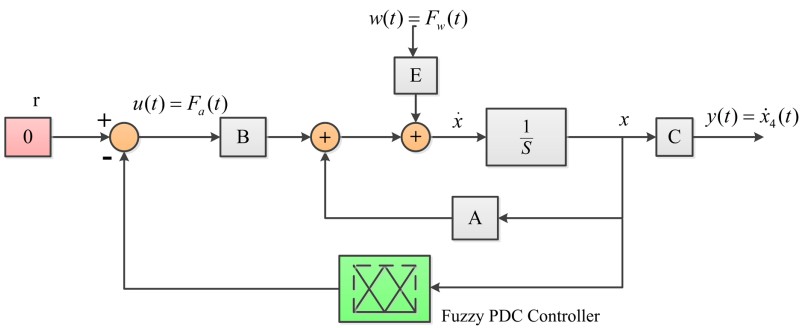

**Figure 4 The closed-loop system with fuzzy PDC controller.**

The PDC technique proposes a method to design a fuzzy controller associated with the fuzzy T-S model Eqs. (7a) and (7b). In PDC, $r$ controllers, usually state feedback, associated with each rule $i$ are designed as

**Control Rule i:**

**IF** $s_1(t)$ is $M_{i1}$ and $\cdots$ and $s_p(t)$ is $M_{ip}$,

**THEN** $u(t) = -F_i x(t), \quad i = 1, 2, \ldots, r.$ (13)

The overall control output of the fuzzy PDC controller is then obtained as

$$u(t) = -\frac{\sum_{i=1}^{r} \alpha_i(s(t))(F_i x(t))}{\sum_{i=1}^{r} \alpha_i(s(t))}$$

$$= -\sum_{i=1}^{r} h_i(s(t))(F_i x(t)),$$ (14)

Note that although in the PDC approach the controllers are local (Eq. (14)), the overall controller scheme should be designed globally to ensure global stability and suitable control performance. By substituting Eq. (14) in (7a) and (7b) the closed loop system can be expressed as

$$\dot{x}(t) = \sum_{i=1}^{r}\sum_{j=1}^{r} h_i(s(t))h_j(s(t))((A_i - B_iF_j)x(t) + E_iw(t)).$$ (15)

Now, by considering a positive definite matrix $P$ and the quadratic Lyapunov function $V$ $(x) = x^T(t)Px(t)$, the closed-loop system (15) is asymptotically stable if the following condition is satisfied (*Tanaka & Wang, 2004*)

$$\{A_i - B_iF_j\}^T\{A_i - B_iF_j\} - P < 0 \quad \forall i, j = 1, 2, \ldots, r,$$ (16)

for $h_i(s(t))\, h_j(s(t)) \neq 0, \forall\, t.$

To design a PDC controller for the hand-glove system, Eqs. (4) and (5), it is required to obtain the T-S fuzzy model of the dynamic system (4). Since our goal is to design an active

controller that guarantees suitable performance for different users, *i.e.* different hand masses $m_1$, it is supposed that $m_1 \in [m_1^{min}, m_1^{max}]$. By defining the premise variable $\psi_h(t)$ as

$$\psi_h(t) := \frac{1}{m_1(t)} \tag{17}$$

and using the definition in Eq. (17) and the possible range of $m_1(t)$, the minimum and maximum range of $\psi_h(t)$ can be obtained as

$$\psi_h^{min} = \min\left\{\frac{1}{m_1(t)}\right\} = \frac{1}{m_1^{max}}, \tag{18a}$$

$$\psi_h^{max} = \max\left\{\frac{1}{m_1(t)}\right\} = \frac{1}{m_1^{min}}. \tag{18b}$$

Then $\psi_h(t)$ can be written in terms of $\psi_h^{min}$ and $\psi_h^{max}$ as follows:

$$\psi_h(t) = M_1(\psi_h(t)) \frac{1}{m_1^{min}} + M_2(\psi_h(t)) \frac{1}{m_1^{max}} \tag{19}$$

where

$$M_1(\psi_h(t)) + M_2(\psi_h(t)) = 1. \tag{20}$$

By using Eqs. (19) and (20), the membership functions can be obtained as follows:

$$M_1(\psi_h(t)) = \frac{\psi_h(t) - \frac{1}{m_1^{max}}}{\frac{1}{m_1^{min}} - \frac{1}{m_1^{max}}}, \tag{21a}$$

$$M_2(\psi_h(t)) = \frac{\frac{1}{m_1^{min}} - \psi_h(t)}{\frac{1}{m_1^{min}} - \frac{1}{m_1^{max}}} \tag{21b}$$

which are demonstrated in Fig. 5. Note that the membership functions $M_1$ and $M_2$ are called "Small" and "Large", respectively.

Then, the hand-glove system model Eq. (4) can be represented as the following fuzzy T-S model for variable hand masses:

**Model Rule 1:**
**IF** $\psi_h(t)$ is "Small",

$$\textbf{THEN} \begin{cases} \dot{x}(t) = A_1 x(t) + B_1 u(t) + E_1 w(t) \\ y(t) = C_1 x(t) \end{cases} \tag{22a}$$

**Model Rule 2:**
**IF** $\psi_h(t)$ is "Large",

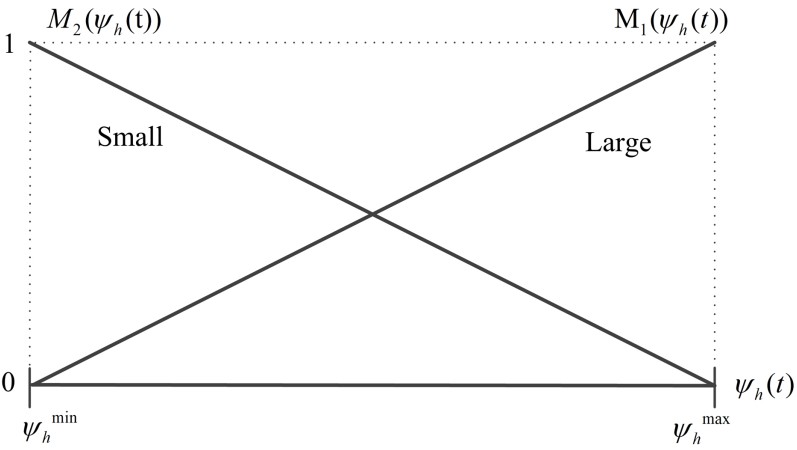

**Figure 5 Membership functions $M_1$ ($\psi_h$(t)) and $M_2$ ($\psi_h$(t)).**

**THEN** $\begin{cases} \dot{x}(t) = A_2 x(t) + B_2 u(t) + E_2 w(t) \\ y(t) = C_2 x(t) \end{cases}$ (22b)

where the fuzzy system's submatrices associated with (22a) and (22b) are obtained as

$$A_1 = \begin{bmatrix} 0 & 0 & 0 & 1 & 0 & 0 \\ 0 & 0 & 0 & 0 & 1 & 0 \\ 0 & 0 & 0 & 0 & 0 & 1 \\ -\dfrac{k_1+k_2}{m_1^{min}} & \dfrac{k_2}{m_1^{min}} & 0 & -\dfrac{c_1+c_2}{m_1^{min}} & \dfrac{c_2}{m_1^{min}} & 0 \\ \dfrac{k_2}{m_2} & -\dfrac{k_2+k_3}{m_2} & \dfrac{k_3}{m_2} & \dfrac{c_2}{m_2} & -\dfrac{c_2+c_3}{m_2} & \dfrac{c_3}{m_2} \\ 0 & \dfrac{k_3}{m_3} & -\dfrac{k_3}{m_3} & 0 & \dfrac{c_3}{m_3} & -\dfrac{c_3}{m_3} \end{bmatrix}$$

$$A_2 = \begin{bmatrix} 0 & 0 & 0 & 1 & 0 & 0 \\ 0 & 0 & 0 & 0 & 1 & 0 \\ 0 & 0 & 0 & 0 & 0 & 1 \\ -\dfrac{k_1+k_2}{m_1^{max}} & \dfrac{k_2}{m_1^{max}} & 0 & -\dfrac{c_1+c_2}{m_1^{max}} & \dfrac{c_2}{m_1^{max}} & 0 \\ \dfrac{k_2}{m_2} & -\dfrac{k_2+k_3}{m_2} & \dfrac{k_3}{m_2} & \dfrac{c_2}{m_2} & -\dfrac{c_2+c_3}{m_2} & \dfrac{c_3}{m_2} \\ 0 & \dfrac{k_3}{m_3} & -\dfrac{k_3}{m_3} & 0 & \dfrac{c_3}{m_3} & -\dfrac{c_3}{m_3} \end{bmatrix}$$

$$B_1 = B_2 = \begin{bmatrix} 0 & 0 & 0 & 0 & \dfrac{1}{m_2} & -\dfrac{1}{m_3} \\ 0 & 0 & 0 & 0 & -\dfrac{1}{m_2} & 0 \end{bmatrix}^T,$$

$$E_1 = E_2 = \begin{bmatrix} 0 & 0 & 0 & 0 & 0 & \dfrac{1}{m_3} \end{bmatrix}^T,$$

$$C_1 = C_2 = \begin{bmatrix} 0 & 0 & 0 & 1 & 0 & 0 \end{bmatrix}$$ (23)

The designed fuzzy controller shares the same input variables and fuzzy sets with the T-S fuzzy model in the IF parts. So, for the fuzzy models given in Eqs. (22a) and (22b), the following controller rules are designed *via* PDC:

*Control Rule 1:*

**IF** $\psi_h(t)$ is "Small",

$$\textbf{THEN } u(t) = -F_1 x(t) \tag{24a}$$

*Control Rule 2:*

**IF** $\psi_h(t)$ is "Large",

$$\textbf{THEN } u(t) = -F_2 x(t) \tag{24b}$$

where $F_1$ and $F_2$ are the feedback gains.

Now, by considering that the disturbance input, *i.e.* vibration from the hand-held tool, has a limited energy it can be assumed that $w \in L_2 [0,\infty)$ and $\|w\|_2^2 \leq w_{max}$. Then, the following control performances are aimed to be obtained:

1. The closed-loop system remains stable.

2. The transmitted vibration to the hand will be reduced significantly, *i.e.* the effect of disturbance input in the system output will be minimized or mathematically

$$\|y\|_2^2 < \gamma \|w\|_2^2 \tag{25}$$

for all $w \neq 0$ where $\gamma$ is a predefined scalar.

3. The designed control input is enforced to a limited value, *i.e.*

$$\|u\|_2 \leq u^{max}. \tag{26}$$

4. The designed controller performs well for different users/hand masses.

**Theorem:** The feedback gains, $F_i$, of the state feedback controller (24a) and (24b) that stabilize the fuzzy T-S system (22a) and (22b) while minimizing the $\gamma$ in (25) and satisfying the control effort constraint (26) can be obtained by solving the following optimization linear matrix inequality (LMI)-based problem as follows:

$$\min_{\Gamma, M_1, \ldots, M_r} \gamma^2$$

$$\text{subject to} \quad \begin{bmatrix} \begin{pmatrix} -\frac{1}{2}\{\Gamma A_i^T - M_j^T B_i^T + A_i\Gamma - B_i M_j \\ +\Gamma A_j^T - M_i^T B_j^T + A_j\Gamma - B_j M_i\} \end{pmatrix} & -\frac{1}{2}(E_i + E_j) & -\frac{1}{2}\Gamma(C_i + C_j)^T \\ -\frac{1}{2}(E_i + E_j)^T & \gamma^2 I & 0 \\ -\frac{1}{2}(C_i + C_j)\Gamma & 0 & I \end{bmatrix} \geq 0,$$

$$\forall i,j = 1, \ldots, r, \ i \leq j, \tag{27a}$$

$$\begin{bmatrix} \Gamma & M_i^T \\ M_i & u_{max}^2 I \end{bmatrix} \geq 0, \qquad \forall i = 1, \ldots, r, \tag{27b}$$

$$\begin{bmatrix} 1 & x(0)^T \\ x(0) & \Gamma \end{bmatrix} \geq 0 \tag{27c}$$

$$\Gamma > 0, \tag{27d}$$

n which $\Gamma = P^{-1}$, where $P$ is a positive definite matrix and $M_i = F_i \Gamma$.

***Proof:***

***Part 1***: Consider the quadratic Lyapunov function $V((x(t))) = x^T(t) P x(t)$, $P > 0$, and $\gamma > 0$ and $u_{max} > 0$ exist. To prove the disturbance rejection LMI (27a), suppose that for the system (7a) and (7b) the following inequality

$$\dot{V}(x(t)) + y^T(t)y(t) - \gamma^2 w^T(t)w(t) \leq 0 \tag{28}$$

is true. By integrating (28), one can write

$$\int_0^T (\dot{V}(x(t)) + y^T(t)y(t) - \gamma^2 w^T(t)w(t))dt \leq 0. \tag{29}$$

Assuming that $x(0) = 0$, then we have

$$V(x(t)) + \int_0^T (y^T(t)y(t) - \gamma^2 w^T(t)w(t))dt \leq 0. \tag{30}$$

Since $V((x(t))) > 0$, it can be obtained from (25) that

$$\frac{\|y(t)\|_2}{\|w(t)\|_2} \leq \gamma. \tag{31}$$

Thus the $L_2$ norm (31) constraint is true for the fuzzy system (7a) and (7b) if the inequality (28) holds.

Now the LMI condition (27a) can be derived from the inequality (28) as follows

$$\dot{x}^T(t) P x(t) + x^T(t) P \dot{x}(t)$$
$$+ \sum_{i=1}^r \sum_{j=1}^r h_i(s(t))h_j(s(t))x^T(t)C_i^T C_j x(t) - \gamma^2 w^T(t)w(t)$$
$$= \sum_{i=1}^r \sum_{j=1}^r h_i(s(t))h_j(s(t))x^T(t)(A_i - B_iF_j)^T Px(t)$$
$$+ \sum_{i=1}^r \sum_{j=1}^r h_i(s(t))h_j(s(t))x^T(t)P(A_i - B_iF_j)x(t)$$
$$+ \sum_{i=1}^r \sum_{j=1}^r h_i(s(t))h_j(s(t))x^T(t)C_i^T C_j x(t) - \gamma^2 w^T(t)w(t)$$
$$+ \sum_{i=1}^r h_i(s(t))x^T(t)E_i^T Px(t) + \sum_{i=1}^r h_i(s(t))x^T(t)PE_i w(t)$$

 

$$= \sum_{i=1}^{r} \sum_{j=1}^{r} h_i(s(t)) h_j(s(t)) \begin{bmatrix} x^T(t) & w^T(t) \end{bmatrix}$$

$$\begin{bmatrix} (A_i - B_i F_j)^T P + P(A_i - B_i F_j) + C_i^T C_j & PE_i \\ E_i^T P & -\gamma^2 I \end{bmatrix} \begin{bmatrix} x(t) \\ w(t) \end{bmatrix} \leq 0. \tag{32}$$

From the inequality (32), it can be obtained

$$\begin{bmatrix} \left( -\sum_{i=1}^{r} \sum_{j=1}^{r} h_i(s(t)) h_j(s(t)) \{ (A_i - B_i F_j)^T P \right. & -P \sum_{i=1}^{r} h_i(s(t)) E_i \\ \left. + P(A_i - B_i F_j) + C_i^T C_j \} \right) & \\ -\sum_{i=1}^{r} h_i(s(t)) E_i^T P & \gamma^2 I \end{bmatrix} \geq 0, \tag{33}$$

which can be decomposed as

$$\begin{bmatrix} \left( -\sum_{i=1}^{r} \sum_{j=1}^{r} h_i(s(t)) h_j(s(t)) \{ & -P \sum_{i=1}^{r} h_i(s(t)) E_i \\ (A_i - B_i F_j)^T P + P(A_i - B_i F_j) \} \right) & \\ -\sum_{i=1}^{r} h_i(s(t)) E_i^T P & \gamma^2 I \end{bmatrix}$$
$$- \begin{bmatrix} \sum_{i=1}^{r} \sum_{j=1}^{r} h_i(s(t)) h_j(s(t)) C_i^T C_j & 0 \\ 0 & 0 \end{bmatrix}$$

$$= \begin{bmatrix} \left( -\sum_{i=1}^{r} \sum_{j=1}^{r} h_i(s(t)) h_j(s(t)) \{ & -P \sum_{i=1}^{r} h_i(s(t)) E_i \\ (A_i - B_i F_j)^T P + P(A_i - B_i F_j) \} \right) & \\ -\sum_{i=1}^{r} h_i(s(t)) E_i^T P & \gamma^2 I \end{bmatrix}$$
$$- \begin{bmatrix} -\sum_{i=1}^{r} h_i(s(t)) C_i^T \\ 0 \end{bmatrix} \begin{bmatrix} \sum_{i=1}^{r} h_i(s(t)) C_i & 0 \end{bmatrix} \geq 0. \tag{34}$$

The inequality condition (34) is equivalent to

$$\begin{bmatrix} \left( -\sum_{i=1}^{r} \sum_{j=1}^{r} h_i(s(t)) h_j(s(t)) & -P \sum_{i=1}^{r} h_i(s(t)) E_i & \sum_{i=1}^{r} h_i(s(t)) C_i^T \\ \{ (A_i - B_i F_j)^T P & & \\ + P(A_i - B_i F_j) \} \right) & & \\ -\sum_{i=1}^{r} h_i(s(t)) E_i^T P & \gamma^2 I & 0 \\ \sum_{i=1}^{r} h_i(s(t)) C_i & 0 & I \end{bmatrix} \geq 0, \tag{35}$$

which can be rewritten as

$$
\sum_{i=1}^{r}\sum_{j=1}^{r} h_i(s(t))h_j(s(t))
$$
$$
\begin{bmatrix}
\begin{pmatrix} -\dfrac{1}{2}\{(A_i - B_iF_j)^T P + P(A_i - B_iF_j) \\ +(A_j - B_jF_i)^T P + P(A_j - B_jF_i)\} \end{pmatrix} & -\dfrac{1}{2}P(E_i + E_j) & \dfrac{1}{2}(C_i + C_i)^T \\
-\dfrac{1}{2}(E_i + E_j)^T P & \gamma^2 I & 0 \\
\dfrac{1}{2}(C_i + C_i) & 0 & I
\end{bmatrix} \geq 0.
\tag{36}
$$

And, eventually, it can be derived from (36) that

$$
\begin{bmatrix}
\begin{pmatrix} -\dfrac{1}{2}\{(A_i - B_iF_j)^T P + P(A_i - B_iF_j) \\ +(A_j - B_jF_i)^T P + P(A_j - B_jF_i)\} \end{pmatrix} & -\dfrac{1}{2}P(E_i + E_j) & \dfrac{1}{2}(C_i + C_j)^T \\
-\dfrac{1}{2}(E_i + E_j)^T P & \gamma^2 I & 0 \\
\dfrac{1}{2}(C_i + C_j) & 0 & I
\end{bmatrix} \geq 0
\tag{37}
$$

Now by multiplying the both side of Eq. (37) by block-diagonal $\{\Gamma\ I\ I\}$, LMI (27a) is obtained, where $\Gamma = P^{-1}$.

***Part 2***: To prove the bounding LMI condition (27b), from $\|u\|_2 \leq u^{max}$ one can write

$$
u^T(t)u(t) = \sum_{i=1}^{r}\sum_{j=1}^{r} h_i(s(t))h_j(s(t))x^T(t)F_i^T F_j x(t) \leq u_{max}^2,
\tag{38}
$$

and then

$$
\frac{1}{u_{max}^2}\sum_{i=1}^{r}\sum_{j=1}^{r} h_i(s(t))h_j(s(t))x^T(t)F_i^T F_j x(t) \leq 1.
\tag{39}
$$

Since $x^T(t)\,\Gamma^{-1}\,x(t) < x^T(0)\,\Gamma^{-1}\,x(0) \leq 1$ for all $t > 0$, the inequality (39) holds if

$$
\frac{1}{u_{max}^2}\sum_{i=1}^{r}\sum_{j=1}^{r} h_i(s(t))h_j(s(t))x^T(t)F_i^T F_j x(t) \leq x^T(t)\Gamma^{-1}x(t),
\tag{40}
$$

and consequently (39) holds. Thus, we have

$$
\sum_{i=1}^{r}\sum_{j=1}^{r} h_i(s(t))h_j(s(t))x^T(t)\left(\frac{1}{u_{max}^2}F_i^T F_j - \Gamma^{-1}\right)x(t) \leq 0.
\tag{41}
$$

The left-hand-side of (41) is equivalent to

$$
\frac{1}{2}\sum_{i=1}^{r}\sum_{j=1}^{r}h_i(s(t))h_j(s(t))x^T(t)\left(\frac{1}{u_{max}^2}F_i^T F_j + \frac{1}{u_{max}^2}F_j^T F_i - 2\Gamma^{-1}\right)x(t)
$$

$$
= \frac{1}{2}\sum_{i=1}^{r}\sum_{j=1}^{r}h_i(s(t))h_j(s(t))x^T(t)
$$

$$
\left(\frac{1}{u_{max}^2}(F_i^T F_j + F_j^T F_i) - \frac{1}{u_{max}^2}(F_i^T - F_j^T)(F_i - F_j) - 2\Gamma^{-1}\right)x(t) \tag{42}
$$

$$
\leq \frac{1}{2}\sum_{i=1}^{r}\sum_{j=1}^{r}h_i(s(t))h_j(s(t))x^T(t)\left(\frac{1}{u_{max}^2}(F_i^T F_j + F_j^T F_i) - 2\Gamma^{-1}\right)x(t)
$$

$$
= \sum_{i=1}^{r}h_i(s(t))x^T(t)\left(\frac{1}{u_{max}^2}F_i^T F_i - \Gamma^{-1}\right)x(t).
$$

Then, the inequality (41) holds if

$$
\frac{1}{u_{max}^2}F_i^T F_i - \Gamma^{-1} \leq 0. \tag{43}
$$

By defining $\mathbf{M}_i = \mathbf{F}_i\,\Gamma$, we have

$$
\frac{1}{u_{max}^2}M_i^T M_i - \Gamma \leq 0. \tag{44}
$$

and using the Schur Complement, the LMI condition (27b) is obtained. **The proof is completed**.

The overall modeling steps and controller design stages are depicted in Fig. 6.

## SIMULATION RESULTS

According to *Rezali & Griffin (2018)*, the mass of the palm and fingers for any person can be estimated as a percentage of its whole body mass as represented in Table 2. Thus, the total mass of the hand can be expressed as $m_1 = 0.006 \times M_{body}$. Here, by assuming that the body mass of a human that works with the hand-held devices may vary between [50, 90] kg, the minimum and maximum possible values of $m_1$ and $\psi_h$ based on Eqs. (18a) and (18b) are calculated and presented in Table 3. In the following, by considering the membership functions given in Eqs. (21a) and (21b) and by substituting the minimum and maximum values of $m_1$ from Table 3, they are obtained as in Eqs. (45a) and (45b) and represented in Fig. 7.

$$
M_1(\psi_h(t)) = 0.68\psi_h(t) - 1.25, \tag{45a}
$$
$$
M_2(\psi_h(t)) = -0.68\psi_h(t) + 2.25 \tag{45b}
$$

The stiffness and damping coefficients of the hand-glove system of Fig. 1 are depicted in Table 4.

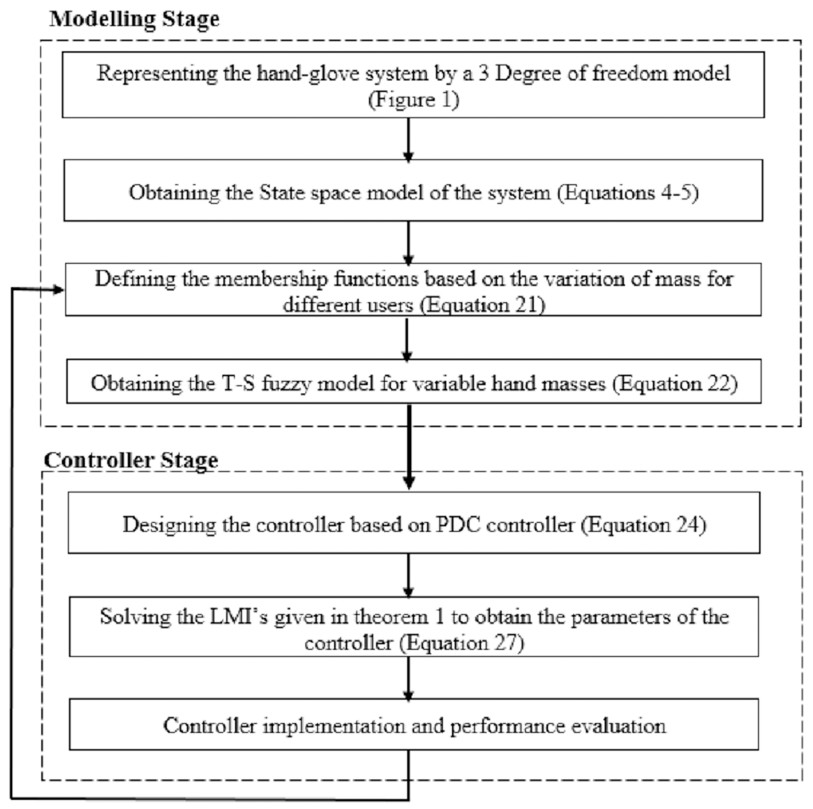

**Figure 6 The overall design procedure.**

**Table 2 Mass of the palm and fingers.**

| Segment | Mass (kg) |
| --- | --- |
| $m_{palm}$ | $0.75 \times 0.006 \times M_{body}$ |
| $m_{fingers}$ | $0.25 \times 0.006 \times M_{body}$ |

**Table 3 Minimum and maximum values of $m_1$ and $\psi_h$.**

| Parameter | Minimum | Maximum |
| --- | --- | --- |
| $m_1$ | 0.3 | 0.54 |
| $\psi_h$ | 1.85 | 3.33 |

## The hand-glove system representation based on the fuzzy T-S model with PDC controller

By considering the T-S fuzzy model (22a) and (22b) and the fuzzy rules (22b) derived for the hand-glove system with variable hand masses and by assuming the maximum and

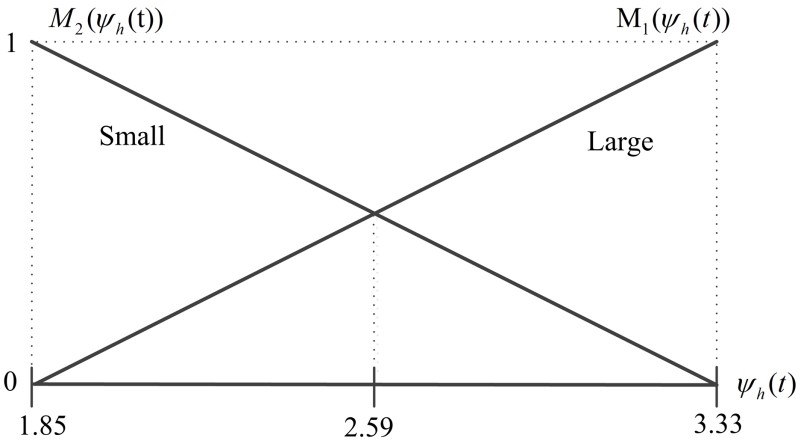

**Figure 7 Membership functions $M_1$ ($\psi_h(t)$) and $M_2$ ($\psi_h(t)$) for $\psi_h^{min}$ = 1.85 and $\psi_h^{max}$ = 3.33.**

**Table 4 The parameter's values used in simulations.**

| Parameter | Value | Parameter | Value |
|---|---|---|---|
| $m_1^{min}$ | 0.3 kg | $c_2$ | 36 N/m |
| $m_1^{max}$ | 0.54 kg | $c_3$ | 1.53 N/m |
| $m_2$ | 0.01 kg | $k_1$ | 0.01 N.s/m |
| $m_3$ | 0.078 kg | $k_2$ | 3667 N.s/m |
| $c_1$ | 1 N/m | $k_3$ | 1152 N.s/m |

minimum hand masses of $m_1^{min} = 0.3\ kg$, $m_1^{max} = 0.54\ kg$, respectively, the associated subsystems based on the Eq. (23) are derived as follows:

$$A_1 = \begin{bmatrix} 0 & 0 & 0 & 1 & 0 & 0 \\ 0 & 0 & 0 & 0 & 1 & 0 \\ 0 & 0 & 0 & 0 & 0 & 1 \\ -12223.37 & 12223.33 & 0 & -123.33 & 120 & 0 \\ 366700 & -481900 & 115200 & 3600 & -3753 & 153 \\ 0 & 2035.33 & -2035.33 & 0 & 2.70 & -2.70 \end{bmatrix}$$

$$A_2 = \begin{bmatrix} 0 & 0 & 0 & 1 & 0 & 0 \\ 0 & 0 & 0 & 0 & 1 & 0 \\ 0 & 0 & 0 & 0 & 0 & 1 \\ -6790.76 & 6790.74 & 0 & -68.52 & 66.67 & 0 \\ 366700 & -481900 & 115200 & 3600 & -3753 & 153 \\ 0 & 2035.33 & -2035.33 & 0 & 2.70 & -2.70 \end{bmatrix} \quad (46)$$

$$B_1 = B_2 = \begin{bmatrix} 0 & 0 & 0 & 0 & 100 & -1.77 \\ 0 & 0 & 0 & 0 & -100 & 0 \end{bmatrix}^T$$

$$E_1 = E_2 = \begin{bmatrix} 0 & 0 & 0 & 0 & 0 & 1.77 \end{bmatrix}^T$$

$$C_1 = C_2 = \begin{bmatrix} 0 & 0 & 0 & 1 & 0 & 0 \end{bmatrix}.$$

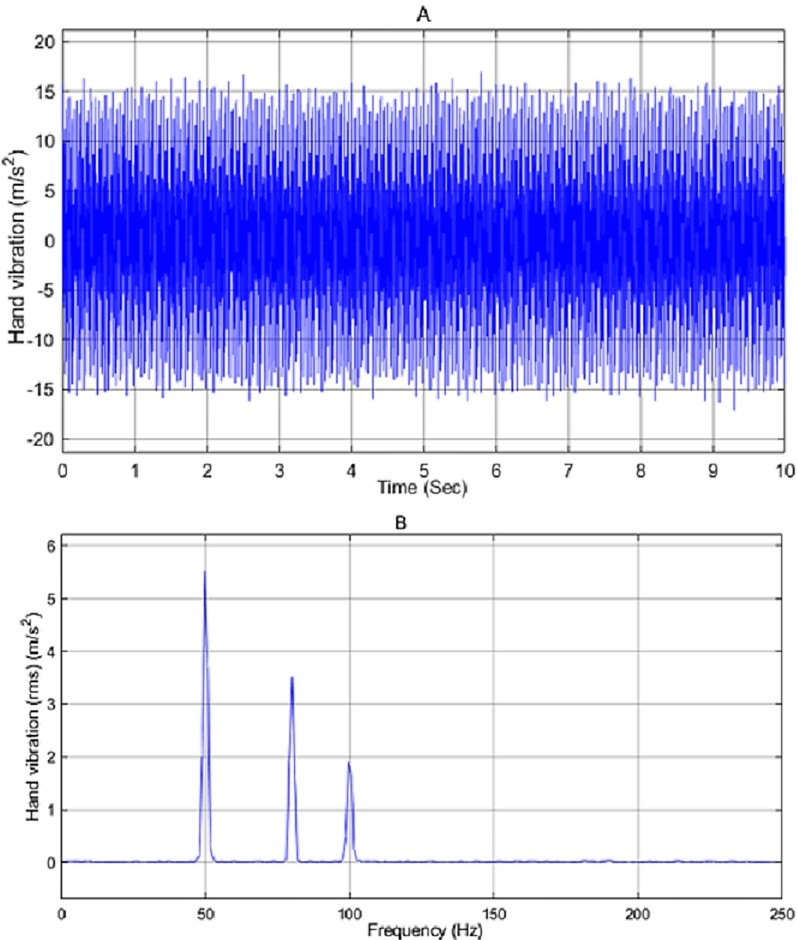

**Figure 8 (A)** Input disturbance to the system and **(B)** frequency spectra of the input vibration.

In the simulation work, the vibration source is assumed to be a 2-stroke engine that is based on *Ko, Ean & Ripin (2011)*, producing a vibration given by the sum of a white noise signal and sine waves with frequencies of 50, 80 and 100 *Hz* and amplitude of 8, 5 and 3, respectively. The input disturbance to the system and its frequency spectrum is shown in Figs. 8A and 8B, respectively. As can be seen in Fig. 8B the engine has the highest peak at 50 *Hz* with a magnitude of 5.5 $m/s^2$ .

The amount of vibration received by the user in the passive mode *i.e.* no controller is applied to the system is represented in Fig. 9. By applying the input disturbance to the system without using any controller, the hand vibration decreased to under 10 $m/s^2$ but still, it is not in the healthy range for humans. As shown in Fig. 9B, the vibration spectra peak related to the hand acceleration in the passive model is reduced to 2.5 $m/s^2$.

## Applying the proposed fuzzy robust PDC controller

In the hand-glove fuzzy system (46), the mass of the hand is considered to be 0.4 kg and values of $m_1^{min}$ and $m_1^{max}$ is chosen to be 0.3 *and* 0.54 *kg*, respectively (Table 3).

**Peer**J Computer Science

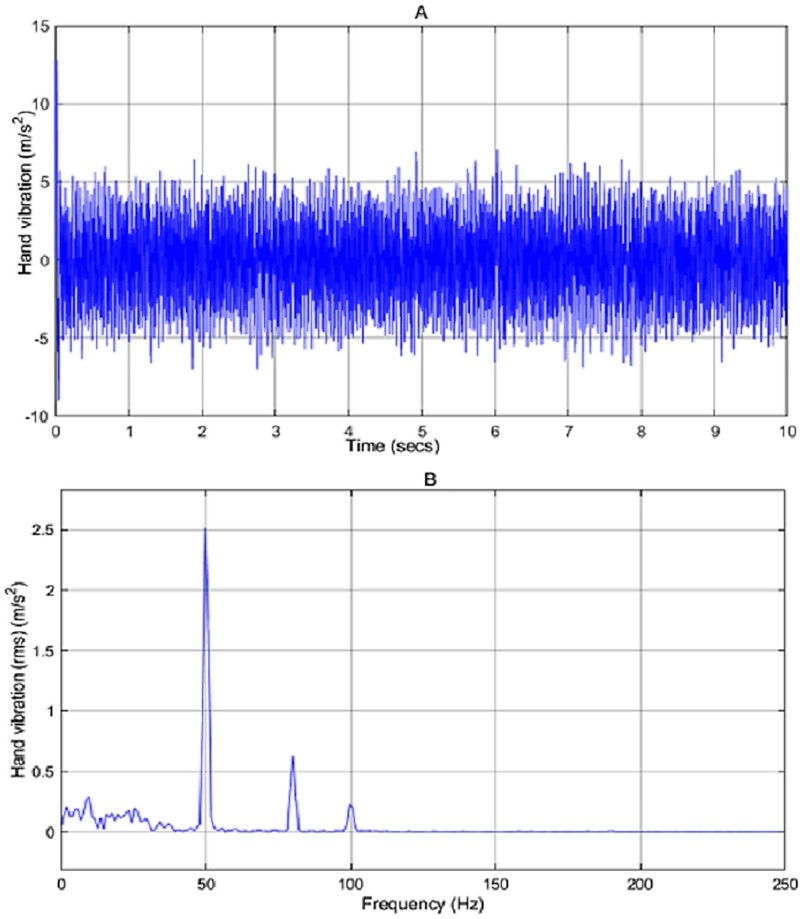

**Figure 9 Vibration of the passive model.** (A) The hand vibration in passive model, without applying a controller, (B) frequency spectra of the hand vibration in passive model.

By considering the system matrices in (46) and solving the LMIs in Theorem 1, which were implemented using MATLAB R2018a and YALMIP (*Lofberg, 2004*) and solved using MOSEK 8, the values of the feedback gains of the fuzzy PDC controller ($F_1$ and $F_2$) and the $P$ matrix are obtained as follows:

$$F_1 = \begin{bmatrix} -7.847 & 9.661 & 4.779 & 25.148 & 572.073 & -0.299 \end{bmatrix},$$
$$F_2 = \begin{bmatrix} -5.412 & 12.267 & 0.208 & 16.006 & 321.739 & -0.407 \end{bmatrix}, \tag{47}$$

$$P = \begin{bmatrix} 14.685 & -7.641 & -2.492 & 1.784 & -0.035 & 0.224 \\ -7.643 & 7.925 & -0.183 & -0.047 & 0.067 & 0.026 \\ -2.492 & -0.183 & 2.052 & -0.393 & 0.006 & 0.045 \\ 1.784 & -0.047 & -0.393 & 0.586 & 0.010 & 0.081 \\ -0.035 & 0.067 & 0.006 & 0.010 & 0.001 & 0.005 \\ 0.224 & 0.026 & 0.045 & 0.081 & 0.005 & 0.040 \end{bmatrix} \tag{48}$$

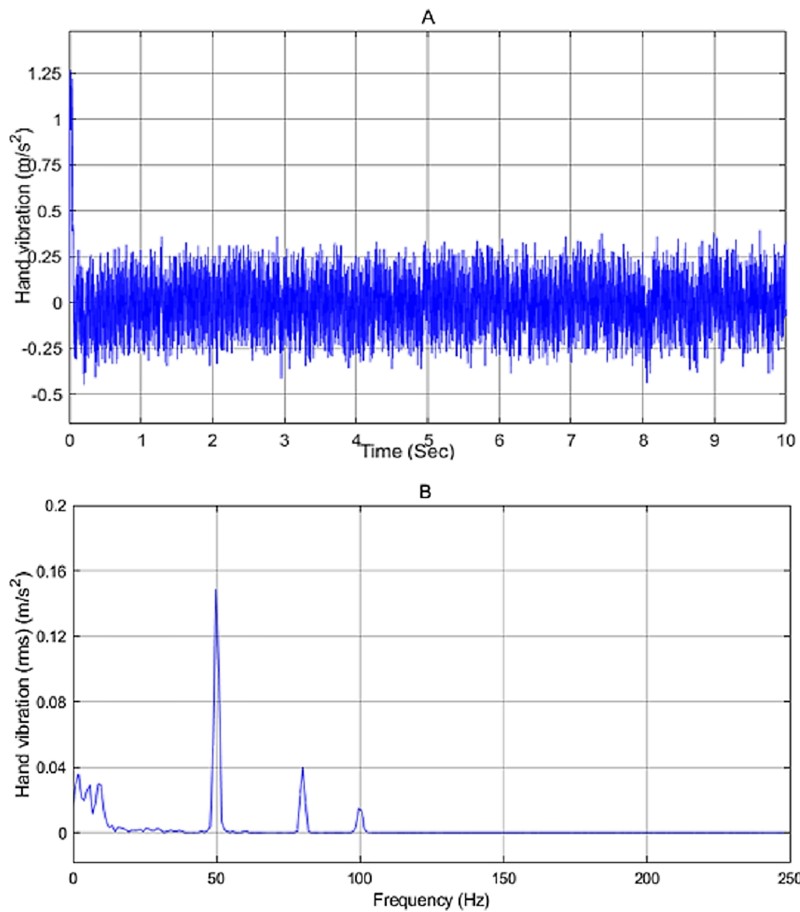

**Figure 10 System vibration with PDC controller.** (A) The vibration at the hand after applying PDC controller; (B) the frequency spectra of the hand vibration after applying PDC controller.

The above feedback gains $F1$ and $F2$ give positive eigenvalues of $P$ that satisfy the positive definiteness of $P$ required in the LMI (27d) of Theorem 1.

By applying the designed fuzzy PDC controller to the hand-glove system (46), the results are as shown in Fig. 10. By comparing the obtained results with the PID and AFC controllers, it can be seen that the designed fuzzy PDC controller has a much better performance. As shown in Fig. 10A, we can see that the proposed fuzzy PDC controller has reduced the vibration experienced by the user to around 0.4 $m/s^2$, well within the healthy zone ($<2.5$ $m/s^2$), defined by the international standards.

One can clearly see the excellent performance of the proposed fuzzy PDC controller in suppressing the vibration compared to the other active vibration controllers as illustrated in Fig. 11. From the simulation results, it can be seen that the proposed controller has a good vibration rejection mechanism.

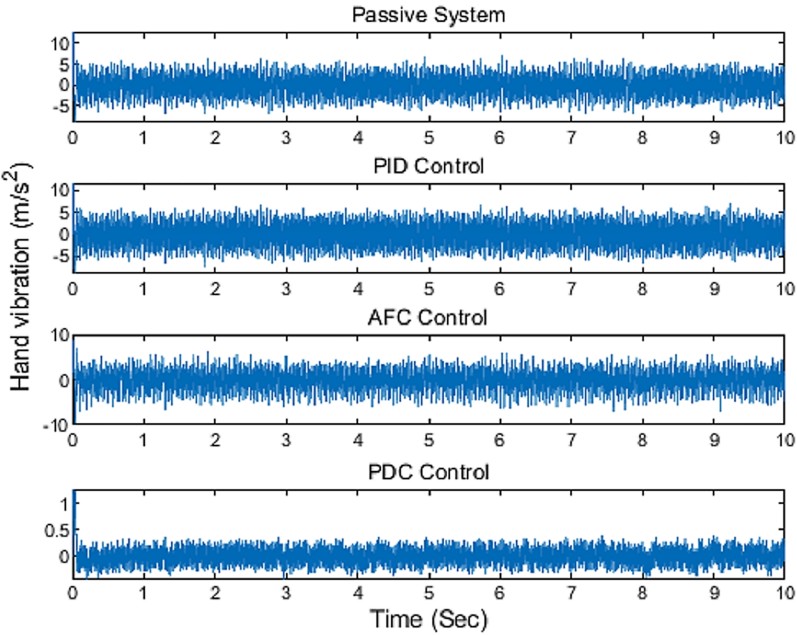

**Figure 11 The comparison result of applying different controllers.**

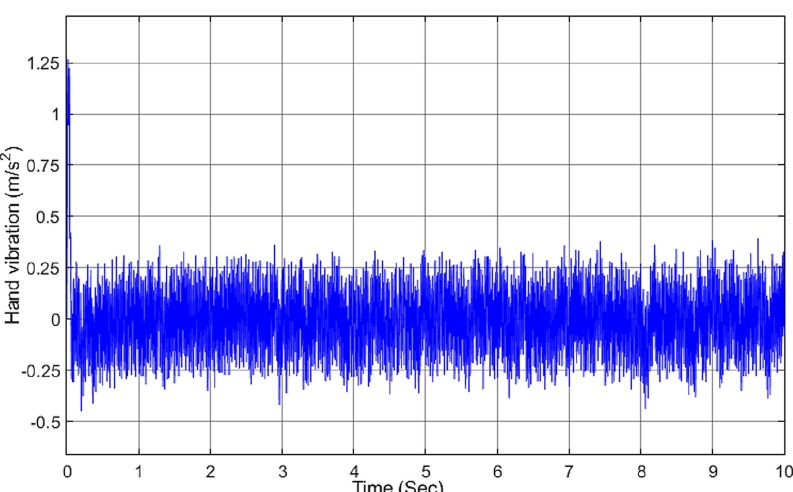

**Figure 12 Vibration of the hand with PDC controller while changing $m_1$.**

To show the robustness of the proposed controller to the variation of the hand masses $m_1$, we have changed the hand mass of hand $m_1$ from 0.4 kg to 0.5 kg. As shown in Fig. 12, the amplitude of the hand vibration has only a small variation when compared with Fig. 10A and is still within the healthy range for humans.

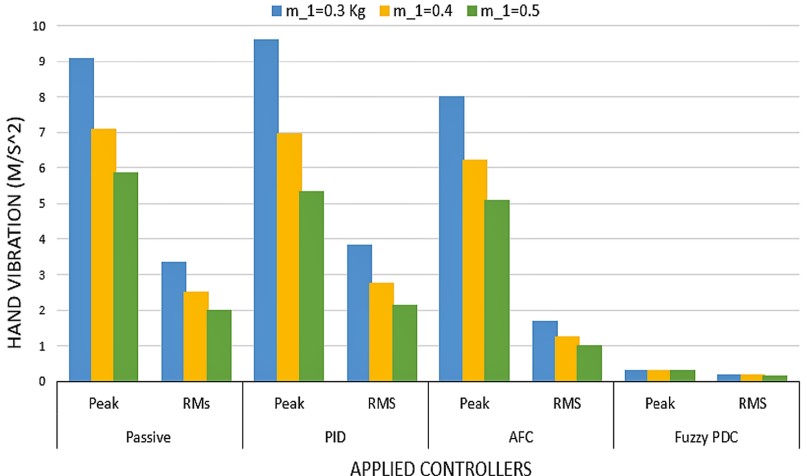

**Figure 13 Hand vibration of varying hand masses and different controllers.**

# CONCLUSION

In this paper, a biodynamic model of the hand-glove system was developed based on the available models for the hand-glove system but focusing more on the vibration experienced by the hand. Then, a robust fuzzy PDC controller was proposed to minimize the vibration transmitted to the hand. By applying the designed controller to the hand-glove system, the amount of vibration experienced by the hand reduced to around 0.4 $m/s^2$ well within a healthy vibration range. Figure 13 summarizes the performances of the different controllers for the hand-glove system experiencing vibrations from vibrating tool usage. The vibrations transferred to the hand using the proposed fuzzy PDC were 93% and 85% less compared to the PID controller and the active force controller (AFC), respectively. Also, the proposed controller was robust to the changes of hand masses.

## Funding

This work was supported by UTM Research University Grant Vote 04G18 and the Ministry of Higher Education Malaysia. The funders had no role in study design, data collection and analysis, decision to publish, or preparation of the manuscript.

## Grant Disclosures

The following grant information was disclosed by the authors:
UTM Research University: 04G18.
 Ministry of Higher Education Malaysia.

## Competing Interests

The authors declare that they have no competing interests.

## Author Contributions

- Leila Rajabpour conceived and designed the experiments, performed the experiments, analyzed the data, performed the computation work, prepared figures and/or tables, authored or reviewed drafts of the paper, and approved the final draft.
- Hazlina Selamat conceived and designed the experiments, performed the experiments, performed the computation work, authored or reviewed drafts of the paper, and approved the final draft.
- Alireza Barzegar conceived and designed the experiments, performed the experiments, analyzed the data, performed the computation work, authored or reviewed drafts of the paper, and approved the final draft.
- Mohamad Fadzli Haniff analyzed the data, authored or reviewed drafts of the paper, and approved the final draft.

## Data Availability

The codes and simulation models are available in the Supplemental Files.

## Supplemental Information

Supplemental information for this article can be found online at http://dx.doi.org/10.7717/peerj-cs.756#supplemental-information.

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
