# Peer review of "Design of a robust active fuzzy parallel distributed compensation anti-vibration controller for a hand-glove system"

_PeerJ Computer Science, doi:10.7717/peerj-cs.756_

## Round 0.1 · original submission · Major Revisions

Now, I have received three reviews and all of them have been noted several technical problems that need to be fixed in a new version of the paper. Please improve the paper according to the reviewers' comments.

·

Basic reporting

Please, see the attached file

Experimental design

please, see the attached file

Validity of the findings

please, see the attached file

Additional comments

please, see the attached file

·

Basic reporting

The English writing is clear and professional, and the structure of the paper is easy to follow. Additionally, the literature references are fair enough.
The raw data is shared, however, there are some problems. There is not any description of how to run the simulations files, I figured it out. Simulation file of AFC controller, PID controller and Passive glove hand system run without any problem. However, the Simulink file of Fuzzy PDC controller fails because variables K_F1 and K_F2 were not declared correctly. I fixed it by changing the variables K_F1 and K_F2 to K_f1 and K_f2 in the file fuzzy_PDC_Controlled_Gloved_Hand_System_Data.m, respectively. I think is important to add a description to run the files and upload the fuzzy_PDC_Controlled_Gloved_Hand_System_Data.m file fixed.

Experimental design

I consider the paper is into the scope of the journal. Moreover, the proposed control scheme is described in detail, and stability proof is also presented.

Validity of the findings

The performance of the proposed fuzzy PDC controller is compared against the PID and AFC controllers. The proposed control scheme proved to be superior in suppressing the vibration experienced by the user. Moreover, the proposed controller is robust to changes of hand masses.

Additional comments

There are some minor problems that need attention. Comment and suggestions are:
a) Equation (2c) is part of Equation (2b). Both Equations should be one.
b) In line 133 of subsection 1.2, should the variables x, u and w be in bold letter?
c) In simulation results, How were the PID gains adjusted? Were they adjusted by experimentation? Some remarks should be added to the paper.
e) In the description of the active vibration control diagram of section 2, the authors comments that the incoming vibration is sensed by using a sensing mechanism, and then a control signal is generated from the measurements of the displacement, velocity or acceleration of the mass M that are fed back. It would be worthy to mention what type of sensors can be used to perform the measurements.
f) The raw data is shared, however, there are some problems. There is not any description of how to run the simulations files, I figured it out. Simulation file of AFC controller, PID controller and Passive glove hand system run without any problem. However, the Simulink file of Fuzzy PDC controller fails because variables K_F1 and K_F2 were not declared correctly. I fixed it by changing the variables K_F1 and K_F2 to K_f1 and K_f2 in the file fuzzy_PDC_Controlled_Gloved_Hand_System_Data.m, respectively. I think is important to add a description to run the files and upload the fuzzy_PDC_Controlled_Gloved_Hand_System_Data.m file fixed.

Reviewer 3 ·

Basic reporting

1- Some explanation can be provided in detail to highlight the contribution.
2- No comparisons are made with the results published in other pertinent references.
3- More details about the selection of the design parameters should be provided for better understand.
4- What is the new of this research?
5- In simulation part, the procedure is too brief to reproduce for interested readers. Some parameters should be given.
6- A general revision of the text is necessary, because there are small typing problems.
7- What are the differences between this work and previous work for controller algorithm?
8- The presentation quality for simulation results should be further improved. Moreover, the physical meaning for the discussion for results should be given, which is very important.
9- The reference list must be updated.

Experimental design

NA

Validity of the findings

NA

---

## Round 0.2 · accepted · Accept

According to the reviewers commenst, this paper can be accepted for publication at PeerJ Computer Science

·

Basic reporting

The English writing is clear and professional, and the structure of the paper is easy to follow. The literature references have been updated by including more recent research. Moreover, the authors have solved the problems with the shared raw data, the raw data have been updated.

Experimental design

I consider the paper is into the scope of the journal. Moreover, the proposed control scheme is described in detail, and stability proof is also presented.

Validity of the findings

The performance of the proposed fuzzy PDC controller is compared against the PID and AFC controllers. The proposed control scheme proved to be superior in suppressing the vibration experienced by the user. Moreover, the proposed controller is robust to changes of hand masses.

Additional comments

The authors have followed the suggestions and comments to improve the manuscript quality. I have no further questions.

Reviewer 3 ·

Basic reporting

The authors made all of my comments. So, I accept the paper for publication in PeerJ CS.

Experimental design

NA

Validity of the findings

NA

Additional comments

NA